# Prognostic Value of Optic Nerve Sheath Diameters after Acute Ischemic Stroke According to Slice Thickness on Computed Tomography

**DOI:** 10.3390/diagnostics14161754

**Published:** 2024-08-12

**Authors:** Han-Bin Lee, Sang Hoon Oh, Jinhee Jang, Jaseong Koo, Hyo Jin Bang, Min Hwan Lee

**Affiliations:** 1Department of Neurology, Seoul St. Mary’s Hospital, College of Medicine, The Catholic University of Korea, Seoul 06591, Republic of Korea; 2Department of Emergency Medicine, Seoul St. Mary’s Hospital, College of Medicine, The Catholic University of Korea, Seoul 06591, Republic of Korea; 3Department of Radiology, Seoul St. Mary’s Hospital, College of Medicine, The Catholic University of Korea, Seoul 06591, Republic of Korea

**Keywords:** acute ischemic stroke, optic nerve sheath diameter, computed tomography, neurological outcome

## Abstract

The optic nerve sheath diameter (ONSD) can predict intracranial pressure and outcomes in neurological disease, but it remains unclear whether a small ONSD can be accurately measured on routine CT images with a slice thickness of approximately 4–5 mm. We measured the ONSD and ONSD/eyeball transverse diameter (ETD) ratio on routine-slice (4 mm) and thin-slice (0.6–0.75 mm) brain CT images from initial scans of acute ischemic stroke (AIS) patients. ONSD-related variables, National Institutes of Health Stroke Scale (NIHSS) scores, and age were compared between good (modified Rankin Scale [mRS] ≤ 2) and poor (mRS > 2) outcomes at discharge. Among 155 patients, 38 had poor outcomes. The thin-slice ONSD was different between outcome groups (*p* = 0.047), while the routine-slice ONSD showed no difference. The area under the curve (AUC) values for the ONSD and ONSD/ETD were 0.58 (95% CI, 0.49–0.66) and 0.58 (95% CI, 0.50–0.66) on the routine-slice CT, and 0.60 (95% CI, 0.52–0.68) and 0.62 (95% CI, 0.54–0.69) on the thin-slice CT. The thin-slice ONSD/ETD ratio correlated with initial NIHSS scores (r = 0.225, *p* = 0.005). After adjusting for NIHSS scores and age, ONSD-related variables were not associated with outcomes, and adding them to a model with NIHSS scores and age did not improve performance (all *p*-values > 0.05). Although ONSD measurements were not an independent outcome predictor, they correlated with stroke severity, and the thin-slice ONSD provided a slightly better prognostic performance than the routine-slice ONSD.

## 1. Introduction

Stroke is the second leading cause of mortality worldwide, with approximately 5.5 million related deaths each year [1]. Additionally, stroke imposes a significant global burden, with its high morbidity resulting in up to 50% of survivors being chronically disabled [1,2,3,4]. Therefore, patients experiencing deficits as a result of acute stroke and their families are worried about their expected outcomes.

The enhancement in clinical outcome assessments would enable the dissemination of higher-quality information to patients and their families and would facilitate stroke management and rehabilitation. The severity of clinical symptoms, which is measured by the National Institutes of Health Stroke Scale (NIHSS) at baseline, and their progression could be valuable predictors of neurological disability outcomes, which are measured using the modified Rankin Scale (mRS) [5]. Imaging findings have also been found to be useful for predicting clinical courses. In patients with acute ischemic stroke (AIS), the infarct volume—which can be measured from diffusion-weighted magnetic resonance imaging (DWI)—serves as an independent predictor of long-term outcomes [6]. However, for patients with a suspected acute stroke in the emergency department (ED), non-contrast brain computed tomography (CT) scans are recommended because they provide the necessary information for acute management decisions [7]. In cases without hemorrhage, an accurate diagnosis of AIS can be made based on clinical presentation and either a negative CT image or a CT image showing early ischemic changes. Although some of patients have non-contrast CT images showing early ischemic changes, it is challenging to detect these findings in all patients.

Recently, measuring the optic nerve sheath diameter (ONSD) has gained clinical interest as a noninvasive, simple method of measuring intracranial pressure (ICP). An increased ICP leads to cerebrospinal fluid (CSF) flow toward the optic nerve subarachnoid space due to communication between the optic nerve subarachnoid space and the chiasmatic cistern [8]. Consequently, evidence has shown that an increase in the ONSD is an early manifestation of elevated ICP [9,10,11]. However, the prognostic value and necessity of measuring the ONSD in CT scans of patients with AIS have not been thoroughly investigated [12,13,14]. Furthermore, it remains unclear whether the optic nerve sheath of AIS patients with a diameter of approximately 4–7 mm can be accurately measured on routine CT images with a slice thickness of approximately 4–5 mm [15,16]. With recent advances in CT technology, thin-slice brain imaging has become possible in most routine clinical CT scans [17]. These high-resolution CT images are useful for identifying subtle skull base conditions and small structures, such as the optic nerve [18].

Our hypothesis is that the ONSD or the ratio of the ONSD to the eyeball transverse diameter (ETD) (ONSD/ETD) could better predict early functional outcomes in AIS patients when they are measured on thin-slice brain CT than when they are measured on routine-slice brain CT images. The primary aim of this study was to evaluate the prognostic performances of ONSD-related variables on a routine- and thin-slice CT in patients presenting with AIS in the ED. The secondary aims were to compare these variables and to explore the utility of them in combination with other clinical variables, including clinical severity scores.

## 2. Materials and Methods

### 2.1. Study Design and Patients

This retrospective observational study was conducted in the ED of a university hospital between 2019 and 2020. All patients over 18 years of age who were hospitalized for AIS, who had AIS on DWI, and who had initial brain CT imaging, including both routine- and thin-slice images, were eligible for inclusion. The electronic charts, brain images and radiologists’ dictated reports were reviewed to identify adult patients who experienced AIS during the study period. Only patients with an increased signal intensity on DWI and decreased apparent diffusion coefficient values were included in the study. Patients with prestroke mRS scores higher than 2, those with unavailable mRS scores on hospital discharge, or those with artifacts on brain CT were excluded. The study was approved by the Institutional Review Board of the Catholic University of Korea, Seoul Saint Mary’s Hospital (KC23RISI0044), and the requirement for consent was waived due to the retrospective nature of the study.

During the study period, our institution had a brain CT scanning protocol for patients suspected of having AIS. Accordingly, the attending emergency physician or senior emergency medicine resident examined these patients and, if hyperacute ischemic stroke was suspected, performed routine-slice non-contrast brain CT, perfusion CT, and CT angiography. However, if the onset-to-door time was too long for reperfusion therapy, only a non-contrast CT protocol, including thin-slice brain CT images, was conducted. All patients received treatment in accordance with local and international AIS care guidelines [6]. Intravenous thrombolysis or intra-arterial reperfusion treatment was provided based on the severity of initial symptoms and the time between symptom onset and presentation in accordance with international AIS guidelines and physician’s decisions. NIHSS scores were evaluated at least twice—at initial presentation and hospital discharge—by a neurologist, and Glasgow Coma Scale (GCS) scores were assessed daily after admission to the stroke unit.

### 2.2. Demographic and Clinical Data Information

We collected the following demographic and clinical data: age, sex, past medical history, clinical signs, symptoms, NIHSS score, GCS score at stroke unit admission, administration of thrombolytic drugs or endovascular treatment, and length of hospital stay. To evaluate functional outcomes, the modified Rankin Scale (mRS) was administered before the AIS event, at initial presentation, and at hospital discharge.

### 2.3. Radiological Data

Sixty-four-channel scanners (Somatom Sensation 64; Siemens Medical Solutions, Erlangen, Germany) were used for all CT studies. The scanning parameters were as follows: 120 kVp, 380 mAs, field of view = 250 × 250 mm, matrix 512 × 512, and a slice thickness 0.6–0.75 mm. The clinical standard axial images were reconstructed with a slice thickness of 4 mm, a standard kernel for soft tissue, and a sharp kernel for the bone structures. In patients who did not undergo additional contrast CT protocols, the thin-slice (0.6–0.75 mm) axial images were reconstructed with a standard kernel.

Two investigators (SHO and HJB) who were blinded to patient outcomes retrospectively measured the ONSDs of all patients, and the average of both values was calculated. The routine-slice ONSD from the 4 mm routine-slice images was measured in accordance with the methods described in previous studies [19]. The routine-slice image was magnified to 300% at a window width of 350 and a level of 40, and the ONSD was measured using an electronic caliper at a distance of 3 mm behind the eyeball (Figure 1). The ETD was measured from retina to retina [19]. The ONSD and ETD were measured bilaterally, and the mean ONSD and ONSD/ETD ratio were calculated. In the thin-slice images, the thin-slice ONSD and thin-slice ONSD/ETD ratio were measured using the same settings (Figure 1) [20].

### 2.4. Outcome Measurement

Based on the mRS scores at hospital discharge, patients were dichotomized into the good outcome group (mRS score from 1–2) and the poor outcome group (mRS score from 3–5).

### 2.5. Statistical Analysis

All categorical variables are expressed as numbers and percentages, and all continuous variables are expressed as the means with standard deviations or medians with interquartile ranges (IQRs). The chi-square test or the Fisher exact test was used to compare categorical variables between groups, and Student’s *t* test or the Mann-Whitney U test was used to compare continuous variables between groups. Interrater reliability between two of the investigators was assessed using intraclass correlation coefficient (ICC). The predictive values of the parameters were evaluated using the receiver operating characteristic (ROC) curve with a 95% confidence interval (CI), which was calculated using an exact binomial method. The ICCs and Pearson correlation coefficients between ONSD-related variables were calculated. To evaluate the association of the ONSD-related variables with outcomes, odds ratios (ORs) and 95% confidence intervals (CIs) were calculated using univariable and multivariable logistic regression analyses. Combination models using ONSD-related variables and clinical variables were created using logistic regression models. Pairwise comparisons of area under the ROC curve (AUC) values were performed using nonparametric tests [21].

All statistical analyses were performed using IBM SPSS version 24 software (IBM, Armonk, NY, USA). All *p*-values were two-tailed, and *p* < 0.05 was considered statistically significant.

## 3. Results

### 3.1. Characteristics of the Study Participants

During the study period, 324 AIS patients presented to the ED (Figure 2). Among these, brain CT scans from 169 patients did not include thin-slice images, 8 patients had poor prestroke mRS scores, and 2 patients had missing mRS scores at discharge. Ultimately, a total of 155 patients met the study criteria. At hospital discharge, 117 (75.5%) patients had a good functional outcome, while 38 (24.5%) patients had a poor functional outcome (Table 1). The poor outcome group was older (68.9 ± 12.8 vs. 75.1 ± 12.1, *p* = 0.009), and their NIHSS score at initial presentation was higher than that of the good functional group (2.0 [IQR, 1.0–4.0] vs. 5.0 [IQR, 2.8–9.3], *p* < 0.001). Reperfusion therapy was used in one patient in each group. The length of hospital stay was longer in the poor outcome group than in the good outcome group (6.0 days [IQR, 5.0–8.0] vs. 12.0 days [IQR, 7.80–24.8], *p* < 0.001).

### 3.2. Comparison of ONSD-Related Variables between the Outcome Groups

The median symptom-to-CT interval was 24.5 h (IQR, 12.3–54.4 h). The interrater reliability of the ONSD measurements between the two investigators exhibited a high ICC (Appendix A). There was a strong positive correlation between the thin-slice ONSD and the routine-slice ONSD (r = 0.864, *p* < 0.001) (Figure 3A), but the thin-slice ONSD (5.66 ± 0.58 mm) was higher (0.63 ± 0.31 mm) than the routine-slice ONSD (5.03 ± 0.61 mm), and the ICC was 0.71 (95% CI, −0.18–0.91). Figure 3B shows that there was a moderately negative correlation of the routine-slice ONSD with the difference between the two measurements (r = −0.333, *p* < 0.001). While there was no significant difference in the routine-slice ONSD between the outcome groups (4.97 ± 0.53 mm vs. 5.21 ± 0.78 mm, *p* = 0.085), the average thin-slice ONSD in the poor outcome group was significantly higher than that in the good outcome group (5.59 ± 0.52 mm vs. 5.85 ± 0.73, *p* = 0.047) (Table 1). The ONSD/ETD ratios on both routine- and thin-slice images were significantly higher in the poor outcome group than in the good outcome group (0.23 ± 0.03 vs. 0.22 ± 0.02, *p* = 0.034; 0.26 ± 0.03 vs. 0.25 ± 0.02, *p* = 0.010, respectively). All ONSD-related variables, especially the thin-slice ONSD/ETD ratio, were positively correlated with the initial NIHSS scores (r = 0.225, *p* = 0.005) (Figure 4). On the other hand, the correlations between ONSD-related variables and age were weaker and nonsignificant (Appendix A). All ONSD-related variables, especially the thin-slice ONSD/ETD ratio, were negatively correlated with GCS scores at admissions (r = −0.227, *p* = 0.004) (Appendix A).

### 3.3. Prognostic Values of Outcome Predictors

The initial NIHSS score and patient age showed a good ability to predict poor outcomes (AUC, 0.76 [95% CI, 0.69–0.83]; 0.65 [95% CI, 0.57–0.72], respectively) (Figure 5A). Among the ONSD-related variables, the thin-slice ONSD (AUC 0.60 [95% CI, 0.52–0.68]) seemed to have a better discriminating ability than the routine-slice ONSD (AUC 0.58 [95% CI, 0.49–0.66] (Figure 5B). The AUCs of the ONSD/ETD ratios on routine- and thin-slice CT images were 0.58 (95% CI, 0.50–0.66) and 0.62 (95% CI, 0.54–0.69), respectively. However, differences between these variables were not significant (all ps > 0.05).

To investigate whether the ONSD-related variables are independently associated with a poor outcome, potential clinical variables, including the initial NIHSS score and age, were examined using various multivariable logistic regression models (Table 2). After controlling for NIHSS score and age, ONSD-related variables were not significant independent predictors of poor outcome (all ps > 0.05). Figure 5C presents the AUCs of the different models with various ONSD-related variables combined with both NIHSS and age. The AUC for predicting a poor outcome significantly increased when NIHSS scores were added to age (AUC 0.81 [95% CI, 0.74–0.87]). However, adding the ONSD-related variables did not significantly improve the AUC for predicting poor outcomes (all ps > 0.05).

## 4. Discussion

In this study, we assessed and compared the value of measuring the ONSD on both routine- and thin-slice CT images for predicting poor functional outcomes (mRS score > 2) at hospital discharge for patients with AIS. Our results revealed differences in the ONSD on thin-slice CT images and the ONSD/ETD ratios on both routine- and thin-slice CT images between outcome groups; on the other hand, there were no between-group differences in the routine-slice ONSD. ONSD-related variables were found to be associated with the initial NIHSS score and GCS score at stroke unit admission. However, the addition of these variables to a model with initial clinical variables (i.e., the NIHSS score and patient age) did not yield any improvement in discriminative performance.

AIS leads to cytotoxic cerebral edema and cellular death due to the dysfunction of sodium-potassium adenosine triphosphatase pumps and blood-brain barrier disruption, potentially causing secondary brain ischemia by elevating ICP. Therefore, the assessment of ICP could provide valuable information in the management of these patients. We found that the ONSD, which can be used to predict ICP, is a significant predictor of the outcome in AIS patients; this finding is consistent with previous results [22,23]. Seyedhosseini et al. measured the ONSD by ocular ultrasonography in 60 patients presenting with acute stroke symptoms and found that an increased ONSD was related to an increased mortality rate, although most of the deceased patients had a hemorrhagic stroke [22]. We demonstrated a positive correlation between the ONSD measured on brain CT scans and NIHSS scores, consistent with earlier ultrasonography-based findings [23].

Measuring the ONSD using ultrasonography offers a dynamic approach to monitoring ICP [24]. While color Doppler may provide more accurate measurements [25], simple measurements using B-mode are more operator-dependent. Recently, several researchers explored the measurement of the ONSD using brain CT images, which is an important imaging technique for evaluating acute stroke symptoms [12,26,27,28]. In our study, we also analyzed the ONSD/ETD ratio, which quantitatively reflects the relationship between the eyeball and optic nerve for ICP monitoring. This variable was measured due to variations in optic nerve size among individuals, as suggested by Vaiman et al. [29]. ONSD/ETD ratios, especially on thin-slice CT images, provided more informative data regarding short-term functional outcomes and were more strongly associated with the initial NIHSS score. Albert et al. examined 1 mm slice CT images and found that the ONSD and ONSD/ETD ratio were significantly higher in the malignant middle cerebral artery (MCA) infarction group on the initial CT scans than in the nonmalignant group, although these variables were not correlated with functional outcomes [27]. Gokcen et al. measured the ONSD among various AIS subtypes and observed that patients with total anterior circulation infarction (ACI) had the highest ONSD, followed by the partial ACI, posterior circulation infarction, and lacunar infarction groups [15]. This finding reflected the potential value of CT-based ONSD measurements for identifying patients with AIS at high risk of mortality and morbidity, such as total ACI.

When we focused on the clinical severity without consideration of the AIS subtype, the association between ONSD-related variables and functional outcomes varied based on CT image thickness. Despite its convenience, the clinical relevance of the ONSD measurement on CT images has been debated by researchers [12,13]. Recently, Kwon et al. compared ONSD measurements obtained from brain CTs with 4 mm and 0.6 mm slice thicknesses as predictors of 6-month neurological outcomes in hypoxic–ischemic encephalopathy (HIE) patients [20]. They concluded that the routine-slice ONSD was non-clinically significant, but the thin-slice ONSD had a higher sensitivity with an acceptable specificity. Interestingly, the smaller the routine-slice ONSD measurement, the greater the disparity between the two methods. This implies that, in conditions with a smaller ONSD than HIE, such as AIS, the discriminating power of the routine-slice ONSD could be further attenuated. To our knowledge, our study was the first to assess the ONSD across different brain CT slice thicknesses in AIS patients. The association of the routine-slice ONSD with the difference between both ONSD measurements was also calculated herein. Thus, routine-slice images might not accurately reflect the true ONSD, and measuring the ONSD using thin-slice images will improve the prognostic performance in AIS patients, even though measurement reliability could improve in patients with increased ICP.

Subtotal or complete MCA infarctions are found in less than 10% of supratentorial infarcts [30,31,32]. They are commonly associated with serious brain swelling, which usually manifests itself between the second and the fifth day after stroke onset [30,31,32]. Although our study enrolled unselected AIS patients, half of these patients underwent brain CT imaging 24 h after symptom onset. Accordingly, our timing of the CT scans could have influenced the results.

This study is the first to evaluate the association between ONSD measurement and neurological outcome by adjusting for other clinical variables. As a single predictor that can be measured in the ED, the initial NIHSS score was important for predicting AIS outcomes; this is also true for age [33]. Interestingly, ONSD-related variables were related to the initial NIHSS score and, after adjusting for the NIHSS score and age, were not independent predictors of functional outcome. Including ONSD-related variables along with these clinical variables did not enhance the prognostic performance of the model for poor outcomes. Nevertheless, considering the limited evidence that can be obtained from the non-contrast brain CT, we recommend measuring the thin-slice ONSD. Thin-slice images can be reconstructed along with routine-slice images from thin-slice raw data, so there is no concern about additional scans or radiation exposure.

This study had several limitations. First, it was conducted retrospectively in a single hospital without a sample size calculation, resulting in a possibility of statistical underpower. Unlike previous studies including only major AIS patients, our study enrolled unselected AIS patients. Considering the relevance of this prognostic tool during the early phase in the ED, where the results of DWI were unavailable, we believed that ONSD values in unselected AIS patients provided valuable information in ED. However, the observed differences were subtle, with ORs exhibiting wide CIs. Further research with a larger sample size is necessary to elucidate the prognostic efficacy of ONSD measurements. Second, our timing of the CT scans could have biased the results. Recent reports have emphasized the value of dynamic changes in the ONSD [12]. Our median time from symptom onset to CT acquisition was relatively late (i.e., approximately 24 h), and the majority of patients with short onset times were excluded because they had undergone other CT protocols, such as CT angiography and CT perfusion, and did not include thin-slice images. Therefore, our results should be interpreted in the context of these limitations. Third, we used the discharge mRS instead of the long-term outcome for outcome assessment. Finally, the measurement of the ONSD on the brain CT and MRI is known to have moderate to high interrater reliability [34,35,36,37], and our ONSD measurements had excellent ICC. However, it is still a subjective process. Considering the recent advancements in artificial intelligence, establishing measurement reliability using automated rater-independent methods is essential [38].

## 5. Conclusions

In patients presenting with a stroke in ED, ONSD-related variables measured on thin-slice brain CTs appeared to offer a better prognostic performance than those measured on routine-slice brain CTs. Although combining these with clinical variables, including initial NIHSS and age, did not improve the ability to predict poor functional outcomes at hospital discharge, in particular, the thin-slice ONSD, as an imaging variable obtained from the non-contrast CT, can be utilized for predicting neurological outcomes.

## Figures and Tables

**Figure 1 diagnostics-14-01754-f001:**
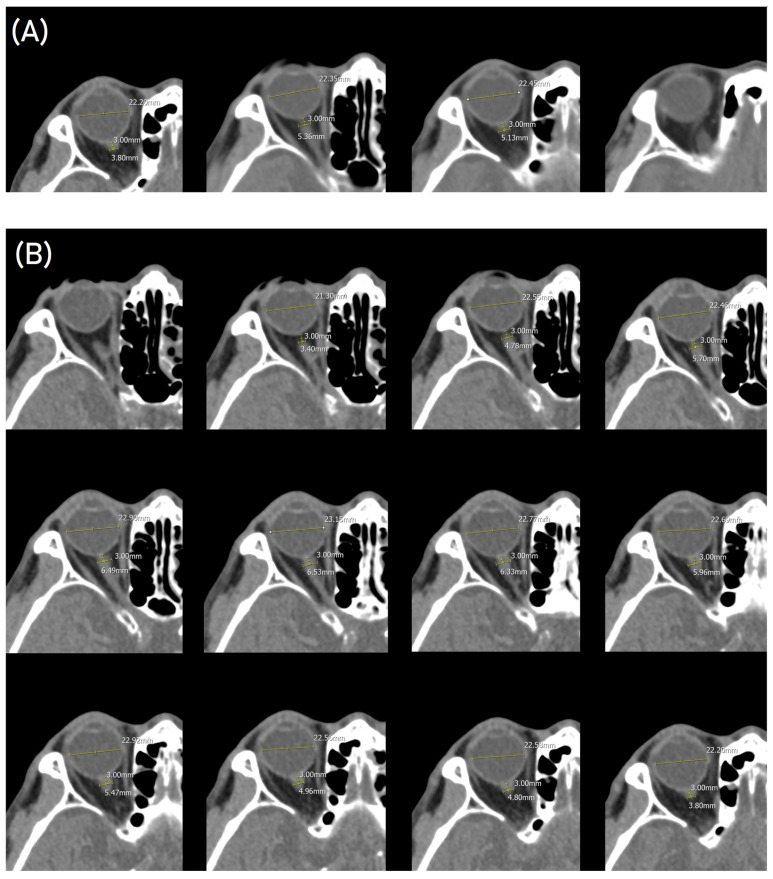
Consecutive right optic nerve axial images of a 78-year-old male with acute ischemic stroke. The optic nerve sheath diameter (ONSD) was measured at 3 mm behind the eyeball. The eyeball transverse diameter (ETD) was measured from retina to retina. (**A**) On 4 mm routine-slice brain computed tomography images, the ONSD could be measured on only 2 axial images, and the routine-slice ONSD and ETD were measured as 5.36 mm and 22.45 mm, respectively. (**B**) On 0.75 mm thin-slice images, the optic nerve sheath was measured in 11 axial images, and the thin-slice ONSD and ETD were measured as 6.53 mm and 23.15 mm, respectively.

**Figure 2 diagnostics-14-01754-f002:**
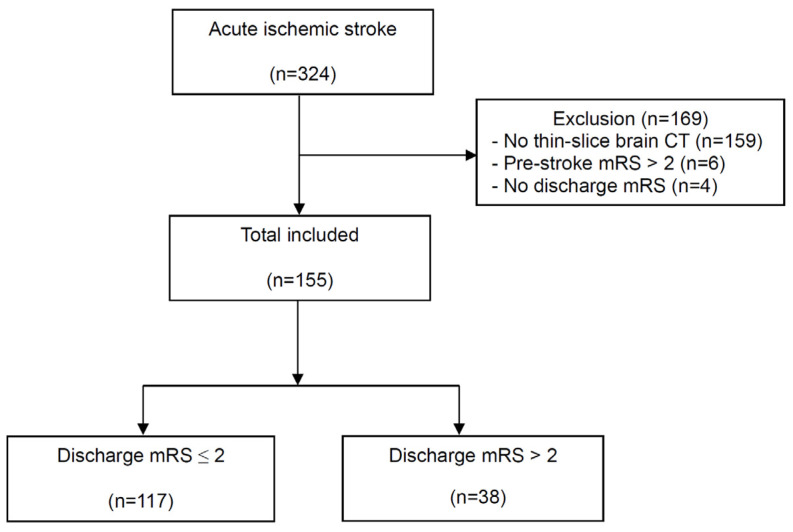
Flow diagram of included patients. mRS, modified Rankin Scale.

**Figure 3 diagnostics-14-01754-f003:**
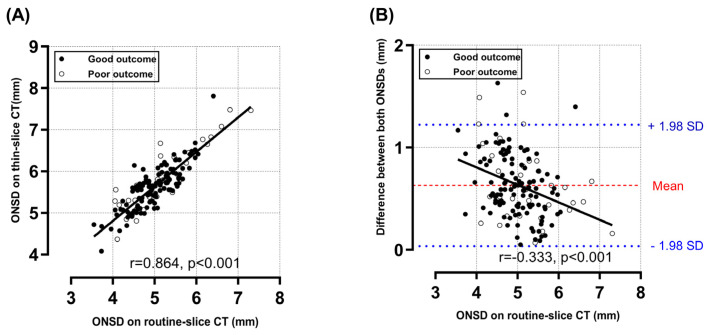
The correlation between the routine-slice ONSD on 4 mm slice CT images and the thin-slice ONSD on 0.60 or 0.75 mm slice CT images. The scatter plot and linear regression line show the correlation between the routine-slice ONSD and thin-slice ONSD (**A**). A Bland-Altman plot indicates that the difference between the thin- and routine-slice ONSD are plotted against the routine-slice ONSD (**B**). The dashed line depicts the mean of the differences; the dotted lines denote the limits of agreement (mean ± 1.96 times SD). The Pearson correlation coefficients (r) and *p*-values are indicated. ONSD, optic nerve sheath diameter; CT, computed tomography; SD, standard deviation.

**Figure 4 diagnostics-14-01754-f004:**
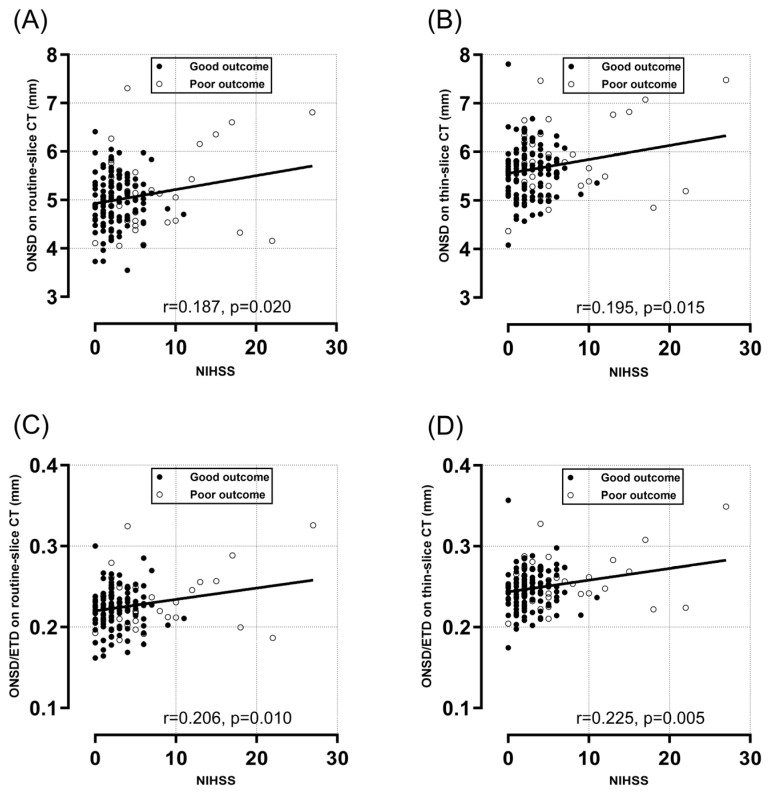
The correlations between the NIHSS score and ONSD-related variables (routine-slice ONSD (**A**), thin-slice ONSD (**B**), routine-slice ONSD/ETD (**C**) and thin-slice ONSD/ETD (**D**)). The Pearson correlation coefficients (r) and *p*-values are indicated. NIHSS, National Institute of Health Stroke Scale; ONSD, optic nerve sheath diameter; CT, computed tomography; ETD, eyeball transverse diameter.

**Figure 5 diagnostics-14-01754-f005:**
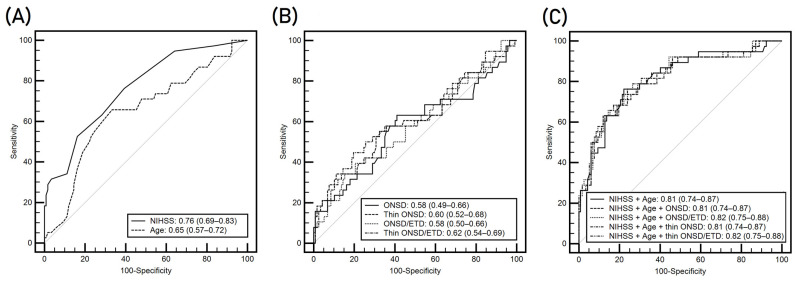
The area under the curves of clinical variables (**A**), ONSD-related variables (**B**), and combination models (**C**) for predicting mRS scores >2 at hospital discharge. Parenthesis indicates the 95% confidence interval. NIHSS, National Institutes of Health Stroke Scale; ONSD, optic nerve sheath diameter; ETD, eyeball transverse diameter.

**Table 1 diagnostics-14-01754-t001:** Baseline characteristics of the included patients.

	mRS Score ≤ 2 (*n* = 117)	mRS Score > 2 (*n* = 38)	*p*-Value
Male	70 (59.8)	19 (50.0)	0.287
Age, years, mean ± SD	68.9 ± 12.8	75.1 ± 12.1	0.009
Comorbidity			
Hypertension	67 (57.3)	25 (65.8)	0.353
Diabetes mellitus	35 (29.9)	11 (28.9)	0.910
Cerebrovascular disease	23 (19.7)	9 (23.7)	0.594
Coronary artery disease	10 (8.5)	3 (7.9)	0.900
Chronic kidney disease	6 (5.1)	1 (2.6)	0.520
Malignancy	9 (7.7)	4 (10.5)	0.584
NIHSS score on initial presentation	2.0 (1.0–4.0)	5.0 (2.8–9.3)	<0.001
mRS score on initial presentation	2.0 (1.0–3.0)	4.0 (3.0–4.0)	<0.001
Glasgow Coma Scale	15.0 (15.0–15.0)	15.0 (13.0–15.0)	<0.001
Symptom onset-to-CT interval, h (IQR)	24.5 (13.4–53.8)	21.6 (5.2–70.6)	0.548
ONSD-related variables			
Routine ONSD, mm	4.97 ± 0.53	5.21 ± 0.78	0.085
Thin ONSD, mm	5.59 ± 0.52	5.85 ± 0.73	0.047
Difference between both ONSDs, mm	0.63 ± 0.30	0.65 ± 0.35	0.739
Routine ETD	22.37 ± 1.08	22.33 ± 0.91	0.827
Thin ETD	22.79 ± 1.08	22.68 ± 0.89	0.570
Routine ONSD/ETD ratio	0.22 ± 0.02	0.23 ± 0.03	0.034
Thin ONSD/ETD ratio	0.25 ± 0.02	0.26 ± 0.03	0.010
Reperfusion therapy	1 (0.9)	1 (2.6)	0.399
NIHSS score at hospital discharge ^a^	1.0 (0.0–2.5)	4.0 (2.0–7.0)	<0.001
mRS score at hospital discharge	1.0 (1.0–2.0)	4.0 (3.0–4.0)	<0.001
Length of hospital stay, day	6.0 (5.0–8.0)	12.0 (7.8–24.8)	<0.001

Data are presented as n (%) for the categorical variables unless otherwise indicated. ^a^ NIHSS scores were assessed in 113 patients with a good outcome and 33 patients with a poor outcome. mRS, modified Rankin Scale; SD, standard deviation; NIHSS, National Institute of Health Stroke Scale; CT, computed tomography; IQR, interquartile range; ONSD, optic nerve sheath diameter; ETD, eyeball transverse diameter.

**Table 2 diagnostics-14-01754-t002:** Results of multivariable logistic regression models for modified Rankin Scale scores >2 at hospital discharge.

	Univariable	Multivariable	Multivariable	Multivariable	Multivariable
	OR (95% CI), *p*	OR (95% CI), *p*	OR (95% CI), *p*	OR (95% CI), *p*	OR (95% CI), *p*
Age, per year	1.04 (1.01–1.08), 0.011	1.05 (1.01–1.09), 0.009	1.05 (1.01–1.09), 0.009	1.05 (1.01–1.09), 0.010	1.05 (1.01–1.09), 0.010
Initial NIHSS score	1.40 (1.20–1.64), <0.001	1.44 (1.22–1.69), <0.001	1.44 (1.22–1.70), <0.001	1.43 (1.22–1.69), <0.001	1.44 (1.22–1.69), <0.001
Routine ONSD, mm	1.90 (1.03–3.48), 0.039	1.62 (0.76–3.45), 0.214	–	–	–
Thin ONSD, mm	2.12 (1.12–4.01), 0.021	–	1.94 (0.88–4.28), 0.102	–	–
Routine ONSD/ETD × 10	4.26 (1.08–16.75), 0.038	–	–	2.61 (0.47–14.51), 0.273	–
Thin ONSD/ETD × 10	6.46 (1.46–28.66), 0.014	–	–	–	4.75 (0.77–29.16), 0.093

OR, odds ratio; CI, confidence interval; NIHSS, National Institute of Health Stroke Scale; ONSD, optic nerve sheath diameter; ETD, eyeball transverse diameter.

## Data Availability

The data presented in this study are available upon request from the corresponding author. The data are not publicly available due to legal restrictions.

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
