# Peer review of "Prognostic Value of Optic Nerve Sheath Diameters after Acute Ischemic Stroke According to Slice Thickness on Computed Tomography"

_diagnostics, 2024, doi:10.3390/diagnostics14161754_

Round 1

Reviewer 1 Report

Comments and Suggestions for Authors

I thank the Editor for giving me the opportunity to correct this valuable manuscript.

I thank the Authors for the interesting topic carried out. The Authors of “Prognostic value of optic nerve sheath diameters after acute ischemic stroke according to slice thickness on computed tomography” have looked at an important aspect concerning neurocritical patients. However, there are some concerns from my humble point of view.  I have tried to highlight some aspects:

1)    Abstract: Is it possible to also indicate the thickness of thin-slice brain imaging, as for routine CT?

2)    Methods, 2.3. Radiological data: why were these patients subjected to CT with normal cut and 64-slice cut? Did they do it with the same machine? Were these patients admitted to peripheral spoke centers? Please, explain better the method on which the study is based, since this bias is probably justified by the context in which the study took place.

3)    Results, 3.1. Characteristics of the study participants: Authors wrote “Reperfusion therapy was used 152 in one patient in each group.” meaning that only one patient out of the 155 recruited was actually subjected to thrombectomy? It seems to me a very strange fact, perhaps justified by a specific bias in the inclusion criteria. Please, explain well.

4)    Results, 3.2. Comparison of ONSD-related variables between the outcome groups: Authors wrote “The median symptom-to-CT interval was 24.5 h (IQR, 12.3–54.4 h)” it is really a very long time for a time-dependent pathology, the element is also repeated in the discussion and then in the Limitations. In my opinion, it should be well explained whether this is normal at your Center or if it is due to the inclusion in this case study (routine and thin slice CT), to then be stressed in the Limitations.

5)    Table 2: why “Routine ONSD/ETD” is × 10 ? I'm very interested in understanding it

6)    Discussion: Honestly, I am perplexed by the analysis of the results and I believe that clarity should be made in the Discussion. Since the ONSD threshold related to high ICP is not defined, it is possible that even if there are numerical differences, these differences do not directly influence the outcome: if there was no real intracranial hypertension, in the acute phase there was no neuro-worsening and therefore the relationship with good or poor outcome remains relative. It would be interesting to understand which of these patients developed malignant edema in the acute phase or lost GCS points...

7)    Discussion: The comparison between ONSD measured in CT and ultrasound is interesting. I would like to point out a study comparing US and MRI, in which objectively there are methodological differences, it could represent a useful citation even if in a different cohort of patients (ref doi: 10.1186/s13089-022-00291-5)

8)    Discussion: Authors wrote “NHSS score” that look like a typo

9)    Discussion: the authors rightly consider a limitation the fact of obtaining a very good ICC in inter-operator reliability: this study found poor intra and inter-rater reliability on MRI and I would say that MRI images in coronal scans can be much more precise than a CT. It would be a pertinent citation, which could also justify the limitations of the results (ref https://doi.org/10.3390/jcm12072713)

I am sorry if some of the observations are not clear enough. I hope that these indications of mine, can help to improve this work.

Reviewer 2 Report

Comments and Suggestions for Authors

Major comments that need to be addressed in a revised manuscript

1.       This well-structured study assessed the prognostic value of optic nerve sheath diameters (ONSD) and ONSD/eyeball transverse diameter (ETD) ratio after acute ischemic stroke in CT images with conventional thickness and thin thickness. However, it is unclear what this study would add to the diagnosis and treatment of acute cerebral ischemia. Discussion in this regard needs to be added.

Specific comments

Abstract

2.       Unfortunately, the objective and conclusion are described rather vaguely.

Materials and Methods

3.       It is strange that the authors did not consider the lesion side and assessed the mean variables of data obtained from both sides. The authors should address this issue.

4.       Unfortunately, the way of measuring optic nerve sheath diameters is unclear. Did the authors perform the measurements on original axial images? If yes, where was the measurement done? Why did the authors not measure the diameter on coronally reconstructed optic nerve images? Please clarify.

Discussion and Conclusion

5.       Overall, do the authors recommend routine-slice CTs or thin-slice CTs to perform? What is the basis of that? Clearly state these issues.

Round 2

Reviewer 2 Report

Comments and Suggestions for Authors

The authors have likely satisfactorily revised according to my comments and suggestions.